# Live cell imaging of single RNA molecules with fluorogenic Mango II arrays

Adam D. Cawte [1,2], Peter J. Unrau [3✉] & David S. Rueda [1,2✉]

RNA molecules play vital roles in many cellular processes. Visualising their dynamics in live cells at single-molecule resolution is essential to elucidate their role in RNA metabolism. RNA aptamers, such as Spinach and Mango, have recently emerged as a powerful background-free technology for live-cell RNA imaging due to their fluorogenic properties upon ligand binding. Here, we report a novel array of Mango II aptamers for RNA imaging in live and fixed cells with high contrast and single-molecule sensitivity. Direct comparison of Mango II and MS2-tdMCP-mCherry dual-labelled mRNAs show marked improvements in signal to noise ratio using the fluorogenic Mango aptamers. Using both coding (β-actin mRNA) and long non-coding (NEAT1) RNAs, we show that the Mango array does not affect cellular localisation. Additionally, we can track single mRNAs for extended time periods, likely due to bleached fluorophore replacement. This property makes the arrays readily compatible with structured illumination super-resolution microscopy.

[1] Single Molecule Imaging Group, MRC London Institute of Medical Sciences, Du Cane Rd, London, UK. [2] Department of Infectious Disease, Faculty of Medicine, Imperial College London, Du Cane Rd, London, UK. [3] Department of Molecular Biology and Biochemistry, Simon Fraser University, 8888 University Drive, Burnaby, BC, Canada. ✉email: punrau@sfu.ca; david.rueda@imperial.ac.uk

In the cell, RNA molecules act as the messenger for genetic information, regulate key RNA processing events and are catalysts in translation and splicing. Imaging the dynamics of single RNA molecules in live cells has been a long standing challenge that if solved holds the promise for routine study of cellular RNAs[1]. For two decades, genetically tagging RNA molecules with MS2 stem loops (MS2-SL) able to bind fluorescent protein (FP)–MS2 coat protein (MCP) fusions have helped visualise individual RNA molecules in live cells during transcription, translation and transport in cell lines, yeast and even mouse models[2–10]. However, the recent observation that MS2 arrays block 5′- and 3′-exonucleases[11,12] and the requirement for nuclear localisation of the FP–MCP protein fusions for enhanced imaging contrast in the cytoplasm places restrictions on the utilisation of this technology. Despite new generations of MS2-SLs (MS2v5 and MS2v6) that reduce effects on RNA processing[13,14], there remains an advantage to developing high contrast imaging strategies that do not rely on fluorescent FP–MCP proteins and that improve imaging contrast within the cell.

Since their development, fluorogenic RNA aptamers have been poised as a potential complementary alternative to the MS2 system[15–24]. Their fluorogenicity enables the induction of fluorescence upon RNA expression and can facilitate high-contrast imaging and offer the potential to use shorter RNA tags. These RNA aptamers after being transcribed, induce fluorescence by binding a fluorogenic dye making it thousands of times brighter upon binding[25]. A wide range of fluorogenic RNA aptamers have been selected and shown to induce aptamer specific fluorescence in live cells. Two of the most promising families of small monomeric aptamers are Spinach and Mango[17–19]. It has long been thought that aptamer arrays, similar to the MS2 cassettes, could enable single-molecule imaging with improved sensitivity. However, both the folding and fluorescence stability of the Spinach aptamer have hindered high-resolution imaging of such arrays[20]. More recently, arrays of the Pepper aptamer have been used to image RNAs with improved resolution[26], but the use of fluorescently labelled proteins is still required for single-molecule detection[27]. The brightness, thermodynamic stability and high-affinity of Mango aptamers bodes well to the accurate detection of RNAs abundance in a cellular environment. The recent isolation of brighter and more photostable Mango aptamers, as well as the detection of as low as five tagged RNAs within an individual foci, suggest the potential to image single molecules of RNA using Mango arrays[28]. Here we show the development of stably folding Mango II repeats and their ability to directly image both coding and non-coding RNAs at single-molecule resolution without affecting their known localisation patterns. By making dual Mango and MS2 arrays of exactly the same size we directly compare the imaging capabilities of the MS2 and Mango systems in a head to head fashion. Furthermore, exchange of the Mango fluorogenic dye in fixed samples extends imaging times, which also benefits super-resolution techniques such as structured illumination microscopy (SIM).

## Results

**Design of a stable Mango array.** We have taken advantage of the recent isolation of brighter and more thermodynamically stable Mango aptamers[28] to develop an RNA imaging array. We chose the Mango II aptamer (Fig. 1a) because of its resistance to formaldehyde fixation, enhanced thermal stability and high affinity for TO1-Biotin (TO1-B)[28,29]. As Mango II requires a closing stem to efficiently fold, we engineered modified stem sequences by altering the Watson–Crick base pairs of three adjacent aptamers to avoid potential misfolding between the aptamer modules (Fig. 1b, sequences in Supplementary Table 1). Bulk fluorescence

intensity measurements of the resulting Mango II triplex array (M2x3) show a fluorescence intensity of ~2.5-fold relative to the Mango II monomer (Fig. 1c). The small intensity decrease likely resulting from homo-FRET. This was further tested by altering the linker length between two aptamers to modulate the relative orientation of each aptamer. In a Mango dimer, a single nucleotide linker is sufficient to fully recover the fluorescence (Supplementary Fig. 1). In a Mango triplex, however, the optimal spacer length for correct folding of the M2x3 construct was found empirically to be 5 nt, potentially due to reduced steric hindrance between monomers (Supplementary Fig. 1d). UV melting curves show identical folding stabilities of the M2x3 array relative to a single monomer (Fig. 1d), a marked improvement compared to a Broccoli triplex aptamer[18] (Supplementary Fig. 1c-h). Similarly, the affinity for the TO1-B fluorophore is maintained in the M2x3 context as shown by measuring their apparent $K_{D,app}$ (Fig. 1e). Increasing the number of M2x3 repeats results in a proportional fluorescence intensity increase (Fig. 1c) showing that folding stabilities and fluorescence efficiencies are maintained in longer arrays. The M2x3 subunit was repeated eight times to directly compare fluorescent intensities with a 24-repeat MS2v5 construct[14]. With the in vitro confirmation of a thermodynamically and fluorescently stable tandem array, we set out to test the fluorescent stability of the tandem array in a cellular environment.

**Single-molecule imaging of Mango array-tagged mRNAs.** To test the Mango array fluorescence in a cellular environment, we chose the mCherry coding sequence (CDS) as a reporter for mRNA transcription (Mango fluorescence) and translation (mCherry fluorescence). An M2x24 array was inserted downstream of the mCherry CDS/stop codon and the construct was placed under the expression of a doxycycline (dox) inducible promoter (Fig. 2a). Cos-7 cells were transfected with plasmid and induced with 1 μg/ml of dox for 1–4 h and either fixed and stained, or imaged in live cells in the presence of TO1-Biotin (200 nM). Upon induction, a significant increase in Mango specific fluorescence could be observed in both fixed and live cells (Fig. 2a). Individual diffraction-limited foci could be observed in both fixed and live cells, indicative of single mRNA molecules. Similarly, the observed foci shared a striking resemblance with the otherwise identical MS2v5x24 array in the presence of tdMCP-EGFP (Fig. 2b and Supplementary Fig. 2a, b). Furthermore, TO1-Biotin specific fluorescence was only apparent with expression of the mCherry-M2x24 mRNA and was absent with a control mCherry-MS2v5x24 mRNA (Fig. 2c).

We determined the average monomer intensity based on step-photobleaching experiments of single foci in fixed cells as described previously[28] (Fig. 2d), and obtained a value of ~40 a.u. in good agreement with our previously published value for the Mango II aptamer[28] (Fig. 2e). The distribution of maximum foci intensities was acquired using FISH-quant[30] using ~7000 foci from 13 cells. The distribution was distinct from the background observed in MS2v5x24 + TO1-B control cells (Fig. 2f). As expected from in vitro data[28], the tdMCP-EGFP labelled foci are on average ~2-fold brighter than the TO1-Biotin dependent foci with both exhibiting a mean diffraction-limited diameter of ~250–300 nm (Supplementary Fig. 2c-e). The M2x24 intensity distribution can be fitted to the sum of three Poisson distributions, which estimates that on average, $7 \pm 1$ ($36 \pm 3\%$), $14 \pm 1$ ($48 \pm 3\%$) and $21 \pm 1$ ($16 \pm 3\%$) of the 24 aptamers in an array are active (Supplementary Fig. 2f). A similar distribution of intensities was observed in live cells, albeit with an ~1.5-fold increase in background fluorescence as shown with the MS2v5x24 + TO1-B control distribution and reduced

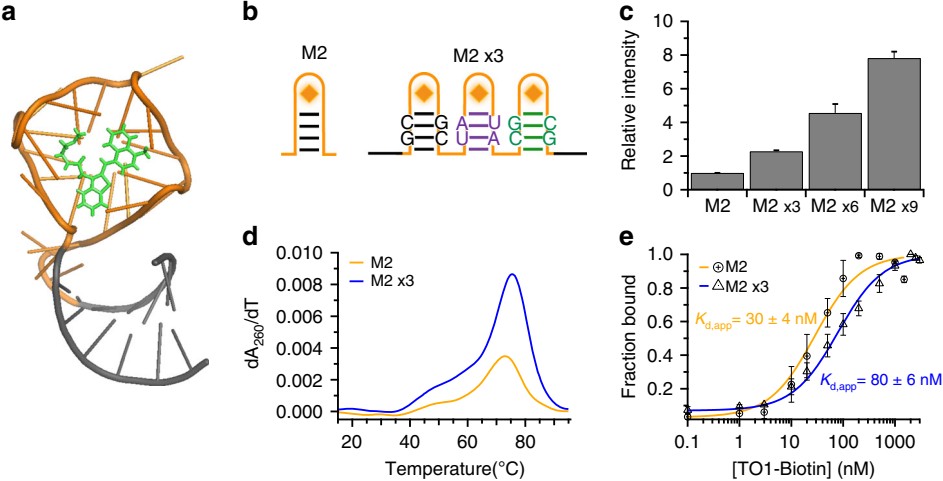

**Fig. 1 In vitro characterisation of tandem Mango II arrays. a** Crystal structure of Mango II aptamer (G-quadruplex—orange, Stem—grey) with TO1-Biotin bound (green). Taken from[29] PDB ID: 6C63. **b** Diagram of Mango II and the Mango II x3 array with mutated bases purple and green. **c** Fluorescence increase of short Mango array constructs (40 nM) upon addition of an excess of TO1-Biotin (1 μM), error bars depict the standard deviation. **d** Differentials of the UV melting curves for each Mango array. **e** Titration of TO1-Biotin for each Mango construct, with a Hill equation fit to determine $K_{d,app}$. Mango II—orange lines and black circles, Mango II x3—blue lines and black triangles.

fluorescence stability of the M2x24 arrays due to the increase in temperature (Fig. 2g, Supplementary Movie 1, 2). The nuclear and cytosolic speed distributions of the M2x24 and MS2v5-EGFPx24 foci in HEK293T cells were comparable (Supplementary Fig. 2g-h and Movie 3).

To further validate the observation of single molecules, we generated a M2/MS2-SLx24 RNA dual label construct encoding adjacent M2x24 and MS2-SLx24 arrays (Fig. 3a). Expression in Cos-7 cells was again induced using dox in the presence of transiently expressed NLS-tdMCP-mCherry. Cells were fixed and stained with 200 nM TO1-Biotin. Nuclear and cytoplasmic foci that were fluorescent in both green and red channels were observed; their size, intensity and localisation varied as expected for transcription sites, nuclear and cytoplasmic mRNAs (Fig. 3a). Removal of either the M2x24 or MS2-SLx24 arrays in either the presence or absence of tdMCP-mCherry confirmed the specificity of each of the fluorescent signals observed (Fig. 3b-d). As expected from the NLS, a bright saturating tdMCP-mCherry signal is observed in the nucleus for the majority of cells (Fig. 3b, d). Other cells showed far less background signal in the nucleus due to limited amounts of tdMCP-mCherry expression (Fig. 3a). This is in stark contrast to the uniform signal-to-noise ratio for both nuclear and cytosolic foci in cells expressing the Mango II array (Fig. 3c), highlighting the power of fluorogenic RNA systems for imaging nuclear RNAs.

Live-cell imaging of the M2/MS2-SLx24 RNA in the presence of both TO1-B and tdMCP-mCherry showed multiple coincident foci per cell (Supplementary Fig. 3a and Supplementary Movie 4), which were fluorescent in both green and red channels. Foci could be readily tracked using basic widefield microscopy and the TrackMate ImageJ plugin[31], with trajectories analysed at 226 ms per frame. Individual trajectories show excellent colocalisation of the foci over long spatial trajectories and times (Fig. 3e, f and Supplementary Movie 5). Plotting the difference between the *xy* position of the Mango and MS2 foci depicts an average difference of ~250 nm with larger fluctuations above the standard deviation only at the highest diffusional speeds and likely resulting from the sequential frame acquisition of the microscope (Fig. 3g). Analysis of mean squared displacement (MSD) values for ~1000 trajectories from multiple cells expressing M2/MS2-SLx24 and labelled with tdMCP-mCherry

show a broad distribution of diffusive speeds (Supplementary Fig. 3b). The increased signal-to-noise in the Mango channel further enhanced the quality of foci detection and length of subsequent tracking (Supplementary Fig. 3c, d and Supplementary Movie 6). Due to the NLS, a strong tdMCP-mCherry signal in the nucleus complicates the analysis of single-molecule trajectories in both the nucleus and cytosol, which requires adjusting the TrackMate plugin[31] thresholds on a cell-by-cell basis, as described in materials and methods. The nuclear foci observed above the background in the mCherry channel (blue distribution) have a slow diffusive behaviour with a mean MSD $= 0.062 \pm 0.019 \; \mu m^2/s$ and a mean intensity ~6-fold greater than that expected for a single mRNA molecule suggesting that they correspond to transcription sites (Supplementary Fig. 3e and Supplementary Movie 7). In contrast the cytosolic foci detected in the mCherry channel have a mean MSD $= 0.464 \pm 0.029 \; \mu m^2/s$ and an intensity distribution with a single peak, both indicative of freely diffusing single molecules. M2/MS2-SLx24 foci detected across the entire cell using TO1-B fluorescence (yellow distribution), show a broader distribution of MSD sharing similarities of both nuclear and cytosolic distributions described previously with a mean MSD $= 0.122 \pm 0.077 \; \mu m^2/s$. Further confirmation of slow diffusing molecules was observed with data acquired at a 3.6 s time frame rate (black distribution) which have a mean MSD $= 0.089 \pm 0.010 \; \mu m^2/s$. As expected, the difference in MSD between M2/MS2-SLx24 and M2x24 arrays imaged in the presence of TO1-B was negligible (Supplementary Fig. 3b, f and Supplementary Movie 4). Quantification of intensities for both M2/MS2-SLx24 foci in live cells shows that both TO1-B and mCherry distributions are distinct from an MS2-SLx24 array in the presence of TO1-B (Supplementary Fig. 3g). The M2/MS2-SLx24 + TO1-B shows a marginally brighter distribution in the Mango channel than the mCherry channel as expected due to mCherry's ~2-fold lower brightness than EGFP and its reduced photostability[32] (Fig. 3h). Quantification of the signal-to-noise ratio of each M2/MS2-SLx24 transcript detected shows a marked increase in the M2x24 + TO1-B channel over the MS2-SLx24 + tdMCP-mCherry channel (Fig. 3i). Taken together, these data show that M2x24 arrays enable the detection and tracking of single mRNA transcripts in live cells and clearly illustrate the benefits in using fluorogenic RNA imaging strategies.

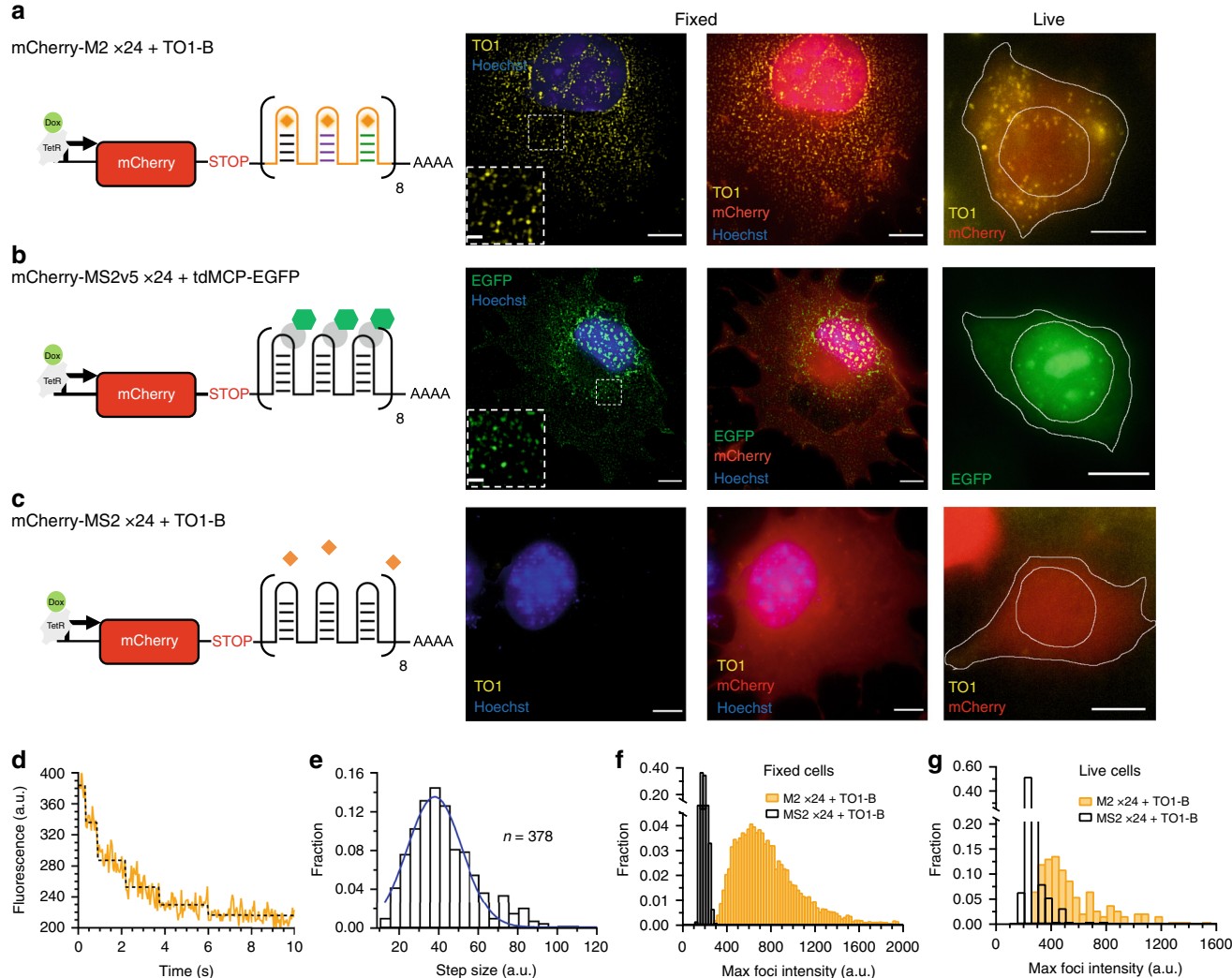

**Fig. 2 Cellular imaging of Mango II arrays. a** Diagram of the mCherry-M2x24 construct + TO1-Biotin controlled by a doxycycline inducible promoter. Followed by representative fixed maximum projections (left & centre) and live cell images of observed cellular foci (right). **b** Diagram of the mCherry-MS2v5x24 construct + tdMCP-EGFP under the control of a doxycycline inducible promoter. Followed by representative fixed maximum projections (left & centre) and live cell images of observed cellular foci (right). **c** Diagram of the mCherry-MS2v5x24 construct + TO1-Biotin under the control of a doxycycline inducible promoter. Followed by representative fixed maximum projections (left & centre) and live cell images of observed cellular foci (right). For (**a**–**c**) TO1-Biotin signal is in yellow, EGFP signal is in green, mCherry is in red, Hoechst 33258 is in blue and scale bars = 10 μm and 1 μm inset. Fixed cell images depict expression in Cos-7 cells and live cell images depict expression in HEK 293 T cells. **d** Representative bleaching curve for an individual foci from a fixed Cos-7 cell, (yellow) fitted with a maximum likelihood step finding algorithm[57] (black). **e** Distribution of 378 calculated bleaching step sizes from 70 individual mCherry-M2x24 foci from eight fixed cells. **f** Fixed cell maximum foci intensity distributions of mCherry-MS2v5x24 (black) and mCherry-M2x24 (yellow) in the presence of 200 nM TO1-Biotin. The M2x24 distribution quantifies~7000 foci from 13 cells. **g** Maximum live-cell foci intensities for cells expressing mCherry-M2x24 (yellow) and mCherry-MS2v5x24 (black) in the presence of 200 nM TO1-Biotin. n = 183 and 187 foci, respectively.

**Mango arrays do not affect localisation of β-actin mRNA.** To test the ability of Mango arrays to recapitulate the localisation pattern of biological mRNAs, we inserted an M2x24 array downstream of the 3′UTR of an N-terminally mAzurite labelled β-actin gene (Fig. 4a). The β-actin 3′UTR contains a zipcode sequence that preferentially localises the mRNA at the edge of the cell or the tips of lamellipodia[33–36]. In addition, we tagged the β-actin coding sequence with a N-terminal Halotag to validate the translation of the β-actin mRNA in fixed cells. Upon transient expression of both tagged β-actin-3′UTR-M2x24 constructs in Cos-7 fibroblast cells, a specific increase in Mango fluorescence could be observed when compared to an equivalent construct containing an MS2v5x24 cassette in the presence of TO1-B (Supplementary Fig. 4a-c). Incubation with the HaloTag-TMR

(Tetramethylrhodamine) ligand gave rise to cells which were efficiently and specifically labelled with TMR. The TMR signal could be seen to accumulate at the periphery of the cells and form cytosolic filaments in both M2x24 and MS2v5x24 labelled mRNAs, confirming the faithful translation and targeting of the Halo-β-actin mRNAs. As expected, the Mango specific signal of the β-actin-3′UTR-M2x24 mRNAs preferentially localise at the edge of the cell or the tips of lamellipodia (Fig. 4b and Supplementary Fig. 4a-c), showing that the array did not affect mRNA transport in either of the constructs. To quantify the mRNA localisation, we calculated the polarisation index (PI) for each localised mRNA, as adapted from a recently reported method[37] (see methods). Both MS2v5-EGFP and M2x24 labelled β-actin mRNAs show a significantly higher PI than β-actin mRNAs

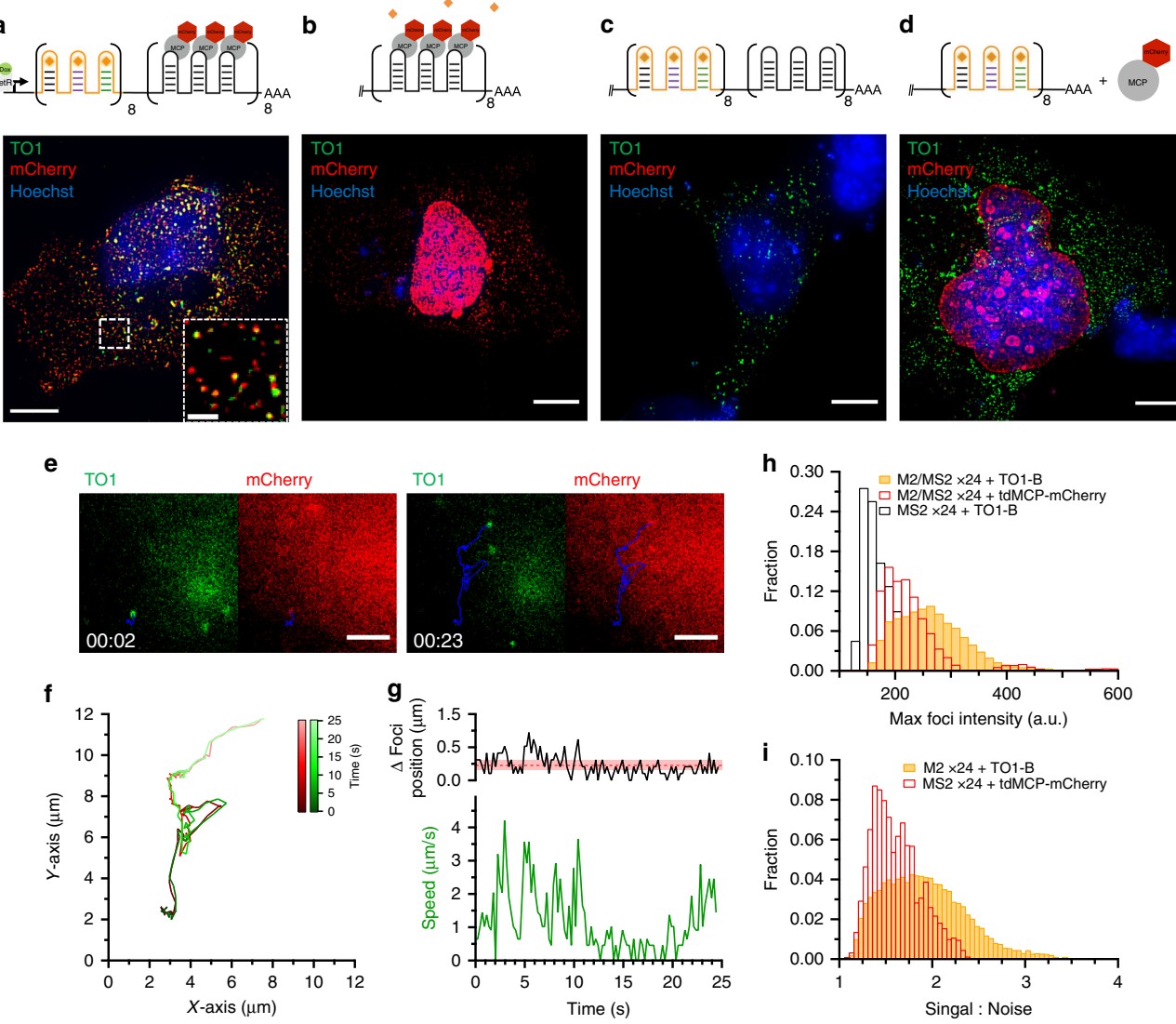

**Fig. 3 Validation of single-molecule imaging with M2/MS2 arrays. a** Diagram and maximum projection of the M2/MS2-SLx24 array under the control of a dox inducible promoter and labelled with TO1-B (200 nM) and NLS-tdMCP-mCherry. Inset shows zoom of selected cytosolic area depicting colocalising signal. Scale bar = 10 μm and 2 μm inset. **b** Diagram and maximum projection of the MS2-SLx24 array in the presence of TO1-B (200 nM) and NLS-tdMCP-mCherry. **c** Diagram and maximum projection of the M2/MS2-SLx24 array in the presence of TO1-B (200 nM). **d** Diagram and maximum projection of the M2x24 array in the presence of TO1-B (200 nM) and NLS-tdMCP-mCherry. **a–d** All cells shown are PFA fixed Cos-7 cells, scale bars = 10 μm unless otherwise stated, TO1-B signal (green), mCherry signal (red) and Hoechst 33258 (blue). **e** Side by side comparison of TO1-B (green) and tdMCP-mCherry (red) labelled trajectories for an M2/MS2-SLx24 single molecule at 2 s and 23 s time points. Trace of the trajectory shown in blue and images taken from (Supplementary Movie 3). Scale bar = 4 μm. **f** XY axis plot of single molecule trajectory from (**e**) showing co-movement of fluorescent signal coloured as a function of time, TO1-B (green) and tdMCP-mCherry (red). **g** Plot of distance between foci localised in the TO1-B and mCherry channels (top) and speed (bottom) against time for the trajectory shown in (**e**) and (**f**). Average distance and standard deviation between foci plotted as shaded red line. **h** Fluorescence intensity distribution of M2/MS2-SLx24 foci from live cell tracking, TO1-B fluorescence (green) mCherry fluorescence (red) MS2-SLx24 + TO1-B background fluorescence (black). $n = 17,544$, 5881 and 33,370 foci, respectively, from 13 cells. **i** Signal-to-noise ratio distribution of foci detected in live cell M2/MS2-SLx24 tracking, TO1-B (green) and mCherry (red), $n = 17,544$ and 5881 respectively from 13 cells.

lacking the 3'UTR and the mCherry-M2x24 construct negative control (Fig. 4b, c), in good agreement with previous results[36] indicating that the Mango array does not interfere with RNA localisation.

To validate the presence of full-length β-actin transcripts the Mango arrays were imaged in conjunction with Stellaris® single-molecule RNA FISH probes against β-actin CDS, mCherry CDS, M2x24 array, or MS2v5x24 array[38,39] (Fig. 4d). Use of these probes validated the presence and accurate detection of full-length β-actin transcripts with either the MS2v5x24 or the M2x24 arrays (Fig. 4e and Supplementary Fig. 4d–f). We quantified the

total number of foci detected in Alexa Fluor™ 488 (AF488) and Quazar® 670 (Q670) channels for probes against the M2 array and mCherry CDS respectively, in cells expressing mCherry-M2x24 ($n_{cell} = 27$). The data shows a positive correlation close to a slope of 1 (0.88) and an $r^2 = 0.7$, indicating that the majority of foci correspond to full-length transcripts (Fig. 4f). Similarly, the percentage of transcripts with overlapping signals in cells expressing β-act-3'UTR-MS2v5x24, β-act-3'UTR-M2x24 and mCherry-M2x24 RNAs was high at ~70% (Fig. 4g). As a control, the β-actin-Q670 probes detecting endogenous β-actin mRNA were used in conjunction with M2-AF488 probes in cells

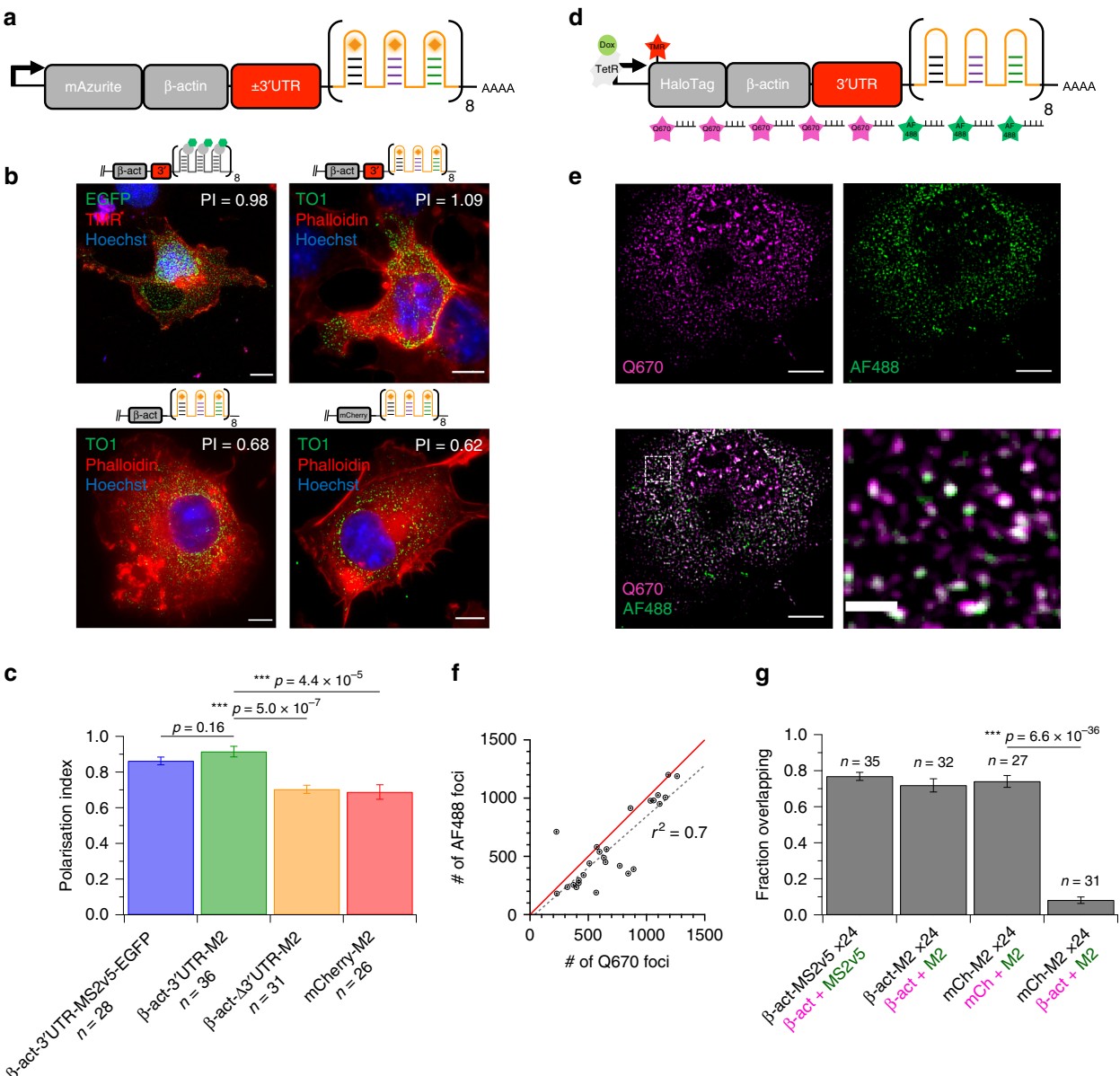

**Fig. 4 Imaging polarisation of β-actin mRNAs in fibroblasts. a** Diagram of the mAzurite labelled β-actin- ± 3′UTR-M2x24 construct under the control of a constitutive promoter. **b** Representative fixed cell images of the localisation patterns for the mAzurite-β-actin-3′UTR-MS2v5x24—EGFP (top left), mAzurite-β-actin-3′UTR-M2x24 (top right) mAzurite-β-actin-Δ3′UTR-M2x24 (bottom left) and mCherry-M2x24 (bottom right) constructs in Cos-7 cells. Stained with TO1-B (200 nM—green channel), TMR or AlexaFluor® 647-Phalloidin (165 nM—red channel) and Hoechst 33258 (1 μg/ml – blue channel), scale bars = 10 μm. Mean PI calculation for RNA localisation within the depicted cell shown in white inset. **c** Mean PI for each Mango tagged construct for multiple cells (*n* indicated), error bars depict the standard error in the mean and p-values calculated using a two-tailed student's T-test. **d** Diagram of Quasar® 670 (Q670 - magenta) and Alexa Fluor™ 488 (AF488—green) smFISH probes against the TMR labelled Halo-β-actin-3′UTR-M2x24 construct under the control of a doxycycline inducible promoter. **e** Maximum projection of a cell expressing Halo-β-actin-3′UTR-M2x24 mRNA and visualised with β-act-Q670 (magenta) and M2-AF488 (green) smFISH probes. Scale bars = 10 μm and 2 μm for high magnification image (lower right panel). **f** Total number of RNA foci in dually labelled Halo-β-actin-3′UTR-M2x24 with smFISH AF488 and Q670 probes using FISH-quant. Single cells foci numbers depicted as grey data points, a line with a gradient = 1 (solid red) and data points fit with a straight line (dashed black—slope = 0.88, $r^2 = 0.7$). **g** Calculated fraction of overlapping signals from multiple dually labelled mRNAs with n as number of cells, error bars depict the standard deviation. RNA construct labelled in black with the corresponding smFISH probes in magenta (Q670) and green (AF488).

expressing the mCherry-M2x24 construct. As expected, this control exhibited a significant decrease in the amount of overlapping signal to ~10% (Fig. 4g and Supplementary Fig. 4g). Taken together these experiments show that the tandem Mango arrays do not affect the cellular localisation of β-actin mRNA and that the tag is efficiently retained within the transcript to a level similar to that of established MS2v5 cassettes.

**Super-resolution RNA imaging with Mango arrays**. Due to the exchange of TO1-B and subsequent fluorescent recovery in fixed cell samples, extended imaging times under pulsed illumination can be achieved[28]. We took advantage of this fluorescence recovery and used structural illumination microscopy (SIM) to acquire Mango tagged RNA images at super-resolution (~100 nm)[40,41]. To first test the recovery of Mango array

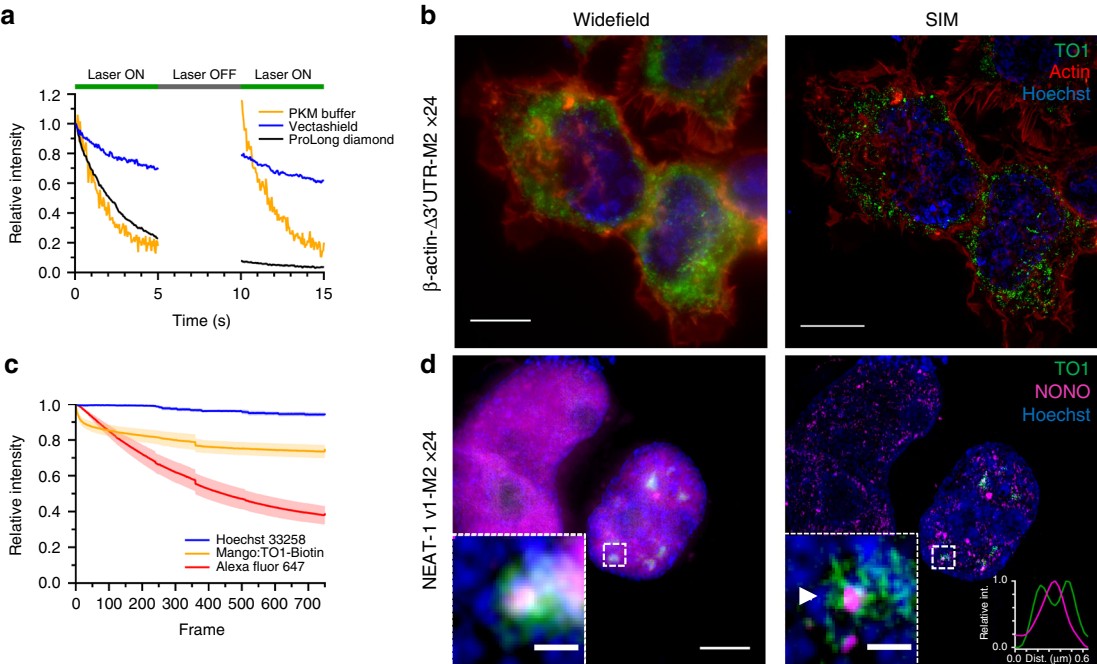

**Fig. 5 Super resolved imaging of Mango tagged RNAs. a** Bleaching curves of individual mCherry-M2x24 foci in fixed HEK293T cells using different media (PKM buffer—yellow, Vectashield® - blue and Prolong® Diamond – black) upon 5 s intervals of constant illumination. **b** Widefield and SIM maximum projections of the β-actin-Δ3'UTR-M2x24 construct (green) costained with Phalloidin-AF647 (red) and Hoechst 33258 (blue), scale bar = 10 μm. HEK293T cells. **c** Mean intensity bleaching curves of Hoechst, Mango and Phalloidin signal during structured illumination acquisition of β-actin-Δ3'UTR-M2x24 expressing HEK293T cells. The standard deviation is plotted as the shaded area around each curve obtained from eight different fields of view. **d** Widefield and SIM maximum projections of NEAT-1 v1-M2x24 construct (green) costained NONO (magenta) and Hoechst 33258 (blue) in HEK293T cells. Zoom of highlighted foci shown as inset. Large scale bars = 5 μm, inset scale bars = 0.5 μm. Inset intensity profile (lower right) of annotated NONO foci (magenta) from SIM maximum projection showing the peripheral NEAT-1 signal (green).

fluorescence, the mCherry-M2x24 construct was imaged with constant wave illumination for 5 s followed by a 5 s pause to allow for fluorophore recovery. Images were acquired in either PKM buffer (10 mM sodium phosphate, 140 mM KCl, 1 mM MgCl₂), Vectashield liquid mounting media or ProLong Diamond hard setting mounting media (Fig. 5a). As expected, we saw efficient recovery of Mango fluorescence in PKM buffer as described previously[28]. Conversely, the hard setting ProLong media showed no fluorescence recovery consistent with fluorophore exchange. Similarly, the Vectashield liquid media reduced fluorescence recovery to an extent that would ultimately lead to the complete loss of Mango signal over the course of a structural illumination acquisition. Therefore, PKM buffer was used in further SIM experiments.

During SIM acquisition, Mango tagged β-actin mRNA was imaged in conjunction with Hoechst 33258 and Phalloidin-AF647 to stain the nucleus and actin cytoskeleton respectively (Fig. 5b). The resulting images reveal well-defined foci below the diffraction limit (~100 nm). SIM reconstruction did not to affect the observed PI as confirmed by using a side by side comparison of widefield and SIM images (Supplementary Fig. 5a–c). Using the SIMcheck ImageJ plugin[42], the total loss of fluorescence during acquisition was measured for multiple cells (Fig. 5c). A mean fluorescence loss of ~20% was seen in the Mango channel over the entirety of image acquisition. The photostable Hoechst dye showed minimal loss at ~5%, whereas the Phalloidin-AF647 signal dropped ~55% over the imaging period. This demonstrates the feasibility of using the stable exchange of RNA Mango TO1-B fluorophores to efficiently reconstruct SIM images—an exciting advance as stable exchange with protein based fluorophore aptamers have not succeeded to date owing to significant photodamage of the protein aptamer tag[43].

Due to the high resolution and enhanced fluorescent contrast obtained within the nucleus, we aimed to image a nuclear retained lncRNA with SIM. To this end, the short isoform of the NEAT-1 lncRNA (NEAT-1v1 - 3.7 kb) was chosen and labelled with the M2x24 array. NEAT-1 is an interesting candidate due to its well-known nuclear coordination of paraspeckles, which has been well characterised using SIM[44]. Upon mild induction, multiple small foci could be observed in the nucleus indicative of nascent transcripts and active transcription. Upon stronger induction, the formation of large Mango specific nuclear foci was observed. Immunostaining for the paraspeckle associated protein NONO (Non-POU domain-containing octamer-binding protein) showed a diffuse colocalization of both signals (Fig. 5d). Upon SIM reconstruction, the NEAT-1-v1 RNA could be seen to surround the core of the paraspeckle containing the NONO protein. Similarly, small microspeckles could be observed outside of the larger paraspeckles. Quantification of the number of NONO foci with adjacent NEAT-1-v1-M2x24 signal showed ~70% of foci had associated lncRNA signal (Supplementary Fig. 5d, e). Given the use of plasmid expression for NEAT-1-v1-M2x24, a full association with paraspeckles was not to be expected. Together, these images are in good agreement with previously published RNA FISH images of the short isoform of NEAT-1 associating with the paraspeckle outer shell and formation of microspeckles[45]. These data demonstrate that the fluorescence recovery of the Mango aptamers can be used to enhance the reconstruction of super-resolved images. Additionally, the improvement in resolution is not solely contained to the use of SIM, but also that the fluorogenic properties of Mango enable higher contrast imaging of RNAs as previously speculated[46] especially within the nucleus where nuclear localisation of FP-MS2 can preclude RNA detection at the single-molecule level.

## Discussion

Since the development of fluorescently tagged bacteriophage coat protein to image RNA molecules in cells[10], single-molecule imaging of mRNAs has provided an important window into the dynamic life of RNA molecules during transcription, splicing, post-transcriptional modification, translation and degradation[2,5–7,47–49]. However, the stability of the capsid bound RNA tag within rapidly degraded mRNAs in yeast has highlighted the need to develop alternative labelling strategies to improve both imaging contrast and reduce aberrant degradation products. In addition to the recently modified MS2-SL sequences[13], fluorogenic RNA aptamers represent an interesting and complementary alternative approach. Fluorogenic RNA aptamers give the added benefit of producing a specific fluorescent signal only in the presence of tagged RNA. To date, fluorogenic RNA aptamers have been used to image short non-coding RNAs with attempts at improving the resolution of imaging with tandem aptamer arrays[20]. However, the fluorescent stability of these arrays both in vitro and in bacterial cells has hindered the observation of single molecules. Recently, two new aptamer-based RNA imaging technologies have emerged both naming their aptamers Pepper. The first using RNA-stabilised fluorescent proteins able to image single mRNAs in live cells[27]. The second, is a fluorogenic RNA aptamer that has been shown to enable RNA imaging in cells, but not at the single molecule level[26]. RNA Mango is likely compatible with these technologies while enabling direct RNA imaging at single molecule resolution without the need for fluorescent proteins.

The results presented here clearly show the improved folding and fluorescent stability of the tandem Mango arrays over previous fluorogenic aptamer designs[20] (Fig. 1 and Supplementary Fig 1). The recent production of brighter, and more stable high-affinity Mango aptamers (in particular Mango II) was instrumental in aiding the efficient production of tandem arrays[28]. These stable tandem arrays enable direct imaging of single RNA molecules in both live and fixed cells. Comparing the single molecule fluorescent intensity distributions of both Mango II and MS2-EGFP labelled mRNAs shows a near identical depiction of diffraction-limited foci, and confirms the expected relative brightness between the two fluorophores measured in vitro[28] (Mango II = ~15,000, EGFP = ~36,000 $M^{-1} cm^{-1}$). Dual labelling individual transcripts with both Mango and MS2 arrays demonstrates clear improvements in signal-to-noise, detection, and tracking for Mango imaging over MS2 imaging, particularly in the nucleus where nuclear localised FP–MCP dominates the signal observed (Figs. 2a, 3c, Supplementary Fig. 3f, g, Supplementary Movie 5, 6). Taken together these observations demonstrate the robust application of the tandem Mango II arrays to image RNAs at single molecule resolution in live cells.

To be broadly applicable, novel labelling strategies need to faithfully maintain an RNAs biological function and accurately represent its localisation pattern. Labelling the well-characterised Halo-tagged-β-actin mRNA[8] with an M2x24 array demonstrated efficient transcription, localisation and translation of the Halo-tagged mRNA, comparable to that of an MS2v5 construct (Supplementary Fig. 4a-c). As expected, no significant perturbation was observed in the preferential localisation of β-actin mRNA at the cell periphery using unbiased analysis[37] of polarisation indices for wild-type and mutant transcripts that lack the 3′UTR zipcode sequence (Fig. 4b, c). Furthermore, quantification of full-length transcripts was also confirmed using smFISH probes for both the β-actin CDS and mCherry CDS (Fig. 4d-g) highlighting the comparable stability of the Mango and MS2 arrays in cells. Due to the absence of nuclear localised FP–MCP the imaging contrast for Mango tagged RNAs in the nucleus was drastically improved, enabling the tagging of nuclear localised RNAs. Labelling of the short NEAT-1-v1 isoform (3.7 kb) with the M2x24 array enabled the detection of nuclear NEAT-1-v1 transcripts, which clearly associated with paraspeckles as expected[44,45]. This fixed cell data suggest Mango arrays may soon enable live-cell imaging of a wider range of nuclear and cytosolic RNAs with improved contrast.

Taking advantage of Mango's extended imaging times in fixed samples due to dye exchange during pulsed illumination, we were able to use SIM to resolve RNA structures below the diffraction limit in fixed cells. Both coding (β-actin) and non-coding RNAs (NEAT-1 v1) could be resolved and demonstrated RNA specific localisation patterns previously observed[36,37,44,45]. This shows the ability of Mango aptamers to be used in conjunction with super-resolution microscopy. Conversely, we hypothesise that by limiting dye concentrations and by controlling pulsed laser illumination, the collection of nm super-resolution images much like existing super-resolution localisation microscopy techniques based on 'blinking'[50] could be enabled.

While the long Mango arrays used in this study allowed us to directly compare with existing MS2 arrays in live cells, there appears to be no fundamental reason to use such long repeat lengths with the Mango system. In this and previous studies[28] we have routinely been able to detect low Mango copy number in cells suggesting that array size minimisation is likely. As widefield microscopy uses relatively intense laser light to image live cells, the photobleaching conditions were quite harsh. Further improvements in live-cell imaging methodology are therefore easily imagined as a consequence. Specifically, further optimisation of fluorophore concentration and Mango array size could allow continuous RNA imaging via fluorophore exchange within living cells. This combined with the use of increasingly standard microscope imaging methodologies such as light sheet[51,52], orbital tracking[53], laser pulse sequence, or laser scanning methodologies[54] appear highly likely to further improve this already compelling approach to the routine imaging of single RNAs within live cells via the use of RNA Mango.

## Methods

**Mango array construction and characterisation.** The conserved stem of the Mango II aptamers was mutated to enable efficient folding of adjacent aptamers. Mango aptamers were mutated by transposing or flipping 6 of the 8 bp in the stem. The first and last G-C pair of the stem were maintained so not to affect the stacking of the G-quadruplex. To create a Mango II x3 construct (M2 x3) 5 nt polyT spacers were used to separate adjacent aptamers and were synthesised as ssDNA by Integrated DNA Technologies (IDT). Similarly, M2 x3 constructs were multi-merized 8 times to obtain a total of 24 aptamers with each triple again being separated by a 5 nt polyT sequence and synthesised by Genewiz Ltd. RNA encoding the tandem Mango arrays was made using the T7 promoter site within the pBSII-SK(-) plasmid to initiate run-off transcription of the MunI linearised plasmid. In vitro transcription used 4 μg of DNA incubated for 4 h at 37°C containing 40 mM Tris-HCl, 30 mM $MgCl_2$, 2 mM spermidine, 1 mM dithiothreitol, 5 mM NTPs, 1 U/μl E. coli inorganic pyrophosphatase (NEB), 4 U/μl T7 RNA polymerase (NEB - pH 7.9) in a total volume of 150 μl. Varied lengths of the transcription products were purified from an 8 M Urea, 18% 29:1 Polyacrylamide: Bis-acrylamide gel run at 20 W for ~4 h in 1X TBE buffer. RNA was extracted from the gel by a crush and soak method using an RNA elution buffer (40 mM Tris-HCl pH 8.0, 0.5 M sodium acetate, 0.1 mM EDTA) followed by ethanol precipitation and re-suspension in nuclease free water.

**Fluorescence measurements of Mango arrays.** Fluorescence emission spectra of each Mango II array were obtained using 40 nM RNA, 200 nM–1 μM TO1-B in sodium phosphate buffer (10 mM sodium phosphate, 140 mM KCl and 1 mM $MgCl_2$, pH 7.2). The modified Thiazole Orange dye (TO1-3PEG-Biotin) was synthesised by Applied Biological Materials Inc. Fluorescence measurements were taken using a Varian Cary Eclipse fluorometer (Agilent) at 22 °C. Mango excitation and emission wavelengths were 505 nm and 535 nm, respectively. TO1-B titrations were conducted by acquiring the peak fluorescence from emission scans at each concentration. Fluorescence measurements were then normalised to the point of saturation, plotted on a log scale (X-axis [dye]) and were fit with a Hill equation (Eq. 1), where x is the concentration of TO1-B, $I_0$ and $I_{max}$ are the normalised minimum and maximum intensities, and $K_{d,app}$ is the apparent dissociation

constant:

$$f(x) = I_0 + (I_{max} - I_0)\frac{x}{x + K_{d,app}}. \tag{1}$$

**UV-melting measurements.** UV measurements were taken using a Varian 100 UV–Vis spectrophotometer and the temperature was controlled with a Peltier cuvette holder and temperature controller. RNA (0.5 μM) was denatured and refolded in 10 mM Sodium Phosphate and 140 mM KCl (pH 7.2) with a temperature ramp of 1 °C/min from 15–95 °C and returning to 15 °C. The differential of the raw data was calculated and was smoothed using a binomial calculation in the Igor Pro 7 software using 9 consecutive data points.

**Mango II array plasmids.** The mammalian expression vectors for the mCherry-M2x24 construct were made by direct digestion of the pUC-57 plasmid synthesised by Genewiz Ltd which, contained an mCherry coding sequence followed by a stop codon and the M2x24 array in the 3'UTR followed by the SV40 poly-adenylation sequence. The doxycycline inducible pLenti-mCherry-M2x24 vector was made by performing QuikChange mutagenesis (Agilent) of the BamHI site in-between the mCherry coding region and Mango array into a PvuI site following the manufacturer's instructions. KOD Hot Start polymerase (Merck Millipore) was used amplify the mCherry-M2x24 sequence to incorporate a 5'' AgeI site and a 3' BamHI site. This product was gel-purified, digested and ligated into a linearised and phosphatase treated pLenti-puro vector (pLenti-dCas9-puro was a gift from Anthony Beucher and Jorge Ferrer—Imperial College). The plasmid is a modified version of the plasmid pCW-Cas9 Addgene plasmid 50661. A pLenti-mCherry-MS2v5x24 plasmid was made similarly by amplification and QuickChange mutagenesis of an mCherry-MS2v5x24 sequence and its insertion into the pLenti backbone using the 5' AgeI and 3' BamHI sites. To image β-actin mRNA the coding region was obtained from a plasmid containing the β-actin and an N-terminal fusion with the BFP derivative mAzurite. The plasmid mAzurite-Actin-7 (a gift from Michael Davidson Addgene plasmid 55227) was modified by incorporating the M2x24 array with 5' XbaI and 3' BclI sites. The minimal β-actin 3'UTR sequence (~500 bp) which encodes the Zipcode sequence was amplified from the plasmid pUBC-HaloTag-bActinCDS-bActinUTR-MS2V5 (a gift from Robert Singer, Addgene plasmid 102718) and cloned into the mAzurite-β-actin-M2x24 plasmid with 5' BamHI and 3' XbaI sites. Similarly, the M2x24 array was cloned into the pUBC-HaloTag-bActinCDS-bActinUTR-MS2V5 using 20 bp of 5' and 3' homology to replace the MS2 cassette using the Hifi builder enzyme mix (NEB). This was later followed by further Gibson assembly of the Halo-bActinCDS-bActinUTR-M2x24 insert into the pLenti-puro backbone using NEB Hifi builder. The short isoform of human NEAT-1 lncRNA (hNEAT-1 v1) was amplified from the plasmid pCRII_TOPO_hNEAT1 (a gift from Archa Fox, Addgene plasmid 61518) and incorporated into an AgeI/BamHI digested pmax-ponA backbone vector, pmax-ponA 12xTRICK - 24xMS2SL was a gift from Jeffrey Chao (Addgene plasmid 64542) which also contained the M2x24 array between the BamHI and MunI sites in replacement of the 24xMS2-SL array. To label the MS2 arrays used in this study, either tdMCP-EGFP or mCherry was expressed from the phage-ubc-nls-ha-tdMCP-gfp plasmid (a gift from Robert Singer - Addgene plasmid 40649) with mCherry incorporated digestion and ligation into XbaI and ClaI sites. Phage-cmv-cfp-24xms2 (a gift from Robert Singer—Addgene plasmid 40651) was also used in intial experiments for imaging MS2-EGFP foci in live cells (Supplementary Movie 1c).

**Cell culture and maintenance.** HEK293T (293T-ATCC® CRL-3216™) and Cos-7 cells (ATCC® CRL-1651™) were grown in Dulbecco Modified Eagle's Medium (DMEM) containing 10% Fetal Bovine Serum (FBS), 2 mM D-Glucose, 2 mM L-Glutamine, 1 mM Sodium Pyruvate and 100 U/ml Penicillin/Streptomycin (Thermo Fisher) and maintained at 37 °C with 5% CO₂ in a humidified incubator. Cells used for imaging were cultured in Ibidi glass bottomed 8-well chamber slides (Ibidi GmbH). Approximately 15–20,000 cells were seeded on to poly-D-Lysine (Sigma) coated dishes and were left to adhere overnight prior to imaging experiments.

**Cell fixation and immunostaining.** Cells were fixed in PKM buffer (10 mM Sodium Phosphate, 140 mM KCl and 1 mM MgCl₂, pH 7.2) containing 4% paraformaldehyde (Thermo Fisher) for 10 min at room temperature followed by permeabilization in 0.2% Triton X-100 for 5–10 min. Cells not requiring immunostaining were washed three times for 5 min each with PKM buffer followed by a 10 min incubation in 200 nM TO1-B diluted in PKM buffer before replacing with imaging media (PKM buffer + 1 μg/ml Hoechst 33258, optional Phalloidin-AF647). For immunostaining, cells were first blocked (2% BSA in PKM) for 30 min followed by primary antibody (1:500 dilutions) incubation for 120 min in blocking solution. Primary antibody Anti-p54-nrb (NONO) (ab50952—5 μg/ml). Secondary antibodies used were Donkey Anti-Rabbit Alexa Fluor 647 (Molecular Probes). Primary antibodies were washed three times for 20 min each in blocking solution followed by incubation with secondary antibody at 1:500 dilution for 60 min, which was subsequently washed as above. After immunostaining, the cells were washed and stained in TO1-B and Hoechst 33258 as described previously. HaloTag

expressing cells were labelled with HaloTag-TMR ligand (Promega-G8251) following the manufacturers instructions for live cell labelling prior to fixation and permeabilization.

**Fluorescence microscopy and live cell imaging.** Live and fixed cell images were taken directly in 8-well chamber slides using a Zeiss Elyra widefield microscope by exciting at 405 nm (Blue), 488 nm (Green), 561 nm (Red) and 642 nm (Far-Red). Emission was detected with a 63× oil immersion objective with a high numerical aperture of 1.4 and band-pass filters at 420–480 nm, 495–550 nm, 570–640 nm and a long-pass filter > 650 nm, respectively. Image acquisition for the fixed samples used 5 mW of power and 200 ms exposure time for each channel, except in photobleaching-assisted microscopy experiments and fast acquisition live-cell experiments (Fig. 3 and Supplementary Fig. 3) where 10 mW laser power and 50 ms exposure time were used. Due to the observation that the Mango signal is stabilised under pulsed illumination in both fixed and live cells, Z-stacks and time series experiments containing more than one colour were acquired by alternating between each channel for an individual frame, leading to recovery and minimal loss of the Mango signal throughout acquisition. For fast frame rate acquisition the laser line was switched between 488 nm and 561 nm with the use of a Zeiss multichannel filter cube (Filter Set 76 HE) to speed up acquisition. An overall frame rate of 226 ms for each colour channel was estimated from the microscope metadata, with the red and green channel acquisition times being 50 ms each. For the slower frame rates used (3.6–5 s), an internal control of acquisition speeds was used in the time-lapse settings.

Plasmids were transfected directly into 8-well chamber slides using Fugene HD following manufacturer's instructions in the culture medium for 16–24 h prior to induction. pLenti-tet-ON plasmids were induced with 1 μg/ml doxycycline for ~2 - 4 h and imaged whereas, pmax-ponA plasmids were induced for ~2–4 h with 1 μM ponasterone A. To facilitate expression of RNA from the pmax-ponA plasmids each construct was cotransfected with the equal amounts of the plasmid pERV-3 which encodes the inducible RXR-VgEcR-VP16 transcription coactivator modules that initiates tight expression of the responsive element encoded in the plasmid upon addition of ponA. Following induction of plasmid expression, the cells were washed once with PKM and either replaced with live-cell imaging media (Fluorobrite DMEM supplemented with 10% FBS, 2 mM D-Glucose, 2 mM L-Glutamine, 1 mM Sodium Pyruvate, 140 mM KCl and 20 mM HEPES, Invitrogen) for longer imaging sessions, or in 20 mM HEPES buffer containing 140 mM KCl and 1 mM MgCl₂ at pH 7.5 for shorter imaging sessions. Both live cell imaging medias were supplemented with 200 nM TO1-B to image Mango tagged RNAs and control constructs. Cells were maintained at 37 °C with 5% CO₂ in a stage top incubator (Tokai Hit).

**Single molecule RNA FISH.** Single molecule RNA FISH samples were prepared using Stellaris® Design Ready probes Human ACTB with Quasar® 670 Dye (VSMF-2003-5), mCherry with Quasar® 670 Dye (VSMF-1031-5) and custom sets of MS2v5 and M2 probes. MS2v5 probes were synthesised by Biosearch™ technologies using their custom probe designer and labelled with Fluorescein dyes (FAM). M2 probes were designed and ordered as HPLC purified AlexaFluor 488™ labelled DNA oligos, synthesised by IDT (Supplementary Table 1). Cells were fixed in PKM buffer (10 mM Sodium Phosphate, 140 mM KCl and 1 mM MgCl₂ pH 7.2) containing 4% paraformaldehyde (Thermo Fisher) for 10 min at room temperature followed by permeabilization in 0.2% Triton X-100 for 5–10 min. FISH probes were hybridised for 16 h at 37˚C directly in ibidi 8-well chamber slides using Stellaris® hybridisation buffer supplemented with 10% deionized formamide. Cells were washed twice at 37 ˚C for 30 min with Stellaris® wash buffer A supplemented with 10% formamide, followed by a 5 min wash with Stellaris® wash buffer B at room temperature. Cells were then imaged in Vectasheild® antifade mounting medium with DAPI (H-1200).

**Structural illumination microscopy.** Structural illumination images were acquired using a Zeiss Elyra microscope. Blue (405 nm) Green (488 nm) and Far Red (642 nm) channels were acquired by sequentially alternating the excitation whilst retaining the same setting with respect to phase, rotation and multi-cube filterset. Images were acquired with 5 phases, 3 rotations and a grating of 32 μm. Images were processed using the automatic reconstruction settings within the Zen software. A TetraSpek™ bead sample was used to create a bead map to which the chromatic aberrations were corrected following manufacturer instructions. The SIMcheck ImageJ plugin was used to determine the quality of the reconstructed images[42], chromatic aberration/drift and most notably the rates of bleaching throughout the acquisition.

**Photobleaching-assisted microscopy.** In order to obtain the appropriate signal-to-noise ratio and time resolution for the analysis of single-step photobleaching, foci were imaged in fixed cells at 20 fps, and 10 mW of widefield laser illumination at 488 nm for ~300 frames. Maximum likelihood estimation was used to determine each of the photobleaching steps within a trace as previously described[55–57]. Due to the fluorescent plateau caused by TO1-B exchange at later time points in the bleaching curves, the step sizes from the initial 100 frames of each bleaching curve

were used. The bleaching steps from 70 single-molecule trajectories were subsequently binned and the histogram was fit with a Gaussian function.

**Image processing, foci detection and particle tracking.** Fixed cell images acquired as z-stacks were processed using a 3D Laplacian of Gaussian (LoG) filter within the FISH-quant Matlab software[30]. Foci detection in fixed samples was limited to foci at least 1.5-fold brighter than background signal and ≤500 nm in diameter. FISH-quant was used to acquire foci intensity values for both positive and negative control constructs. To obtain the distribution of intensity in the MS2x24 control in the presence of TO1-B, the intensity threshold for foci detection was lowered to detect the overall background fluorescence of the cell. The TrackMate ImageJ plugin[31] was used for single particle tracking in live cells. Similar to FISH-quant in fixed samples, a LoG filter was used to enhance the ability to detect single particles in live cells. Foci ~350 nm in diameter with a signal-to-noise ratio ≥ 1 were detected. Frame rates were taken directly from the metadata as compiled in the Zeiss software and ranged from 226 ms–3.6 s. To alleviate some of the non-specific particle tracking caused by excessive nuclear signal from the NLS-tdMCP-mCherry protein, the cytosolic mCherry signal was filtered to leave the lower population of signal based on total foci intensity. To obtain the population of bright nuclear foci in the mCherry channel, an opposing filtering method was used based on total foci intensity and foci signal-to-noise ratio to obtain the high intensity foci. Due to the absence of intense nuclear signal using M2x24 + TO1-B, this data was not filtered based on total foci intensity and both nuclear and cytosolic foci detected and tracked. Following filtering the observed foci tracks were built given a maximum of 2 μm for linking frame to frame displacement and gap-closing. Similarly, a maximum frame gap of 2 was used to recover the partial loss of significant trajectories. Trajectories were finally subjected to a final step of thresholding based on the mean track quality being >30 and each trace having ≥5 localisations. The.csv data output from TrackMate was used to calculate the maximum foci intensity, signal-to-noise ratio and mean square displacement as computed using Matlab.

**Polarisation index and overlapping signal calculations.** The polarisation index (PI) was calculated with a similar method as in described in Park et al.[37] In this publication, they used Eq. 2 to calculate the PI given the mean centroid for both the RNA localisations and the cell:

$$\mathrm{PI} = \frac{\sqrt{(x_{\mathrm{RNA}} - x_{\mathrm{cell}})^2 + (y_{\mathrm{RNA}} - y_{\mathrm{cell}})^2 + (z_{\mathrm{RNA}} - z_{\mathrm{cell}})^2}}{\mathrm{Rg}_{\mathrm{cell}}}. \quad (2)$$

Modifications were made to this process using a custom MATLAB image processing pipeline, in which individual polarisation indices were computed for each observed RNA localisation and its relative distance from the centroid of the cell. The centroid of the cell ($x_{\mathrm{cell}}, y_{\mathrm{cell}}, z_{\mathrm{cell}}$) was computed from the $xy$ coordinates from a given binary mask of the cell and the mid-point of the $z$-axis range ($z_{\mathrm{cell}}$). The $xyz$ coordinates of each RNA localisation ($x_{\mathrm{RNA}}, y_{\mathrm{RNA}}, z_{\mathrm{RNA}}$) was obtained using the ImageJ plugin FociPicker3D using a threshold above the observed cellular background. The PI for a given localisation was then normalised for cell size by dividing by the radius of gyration of the cell ($\mathrm{Rg}_{\mathrm{cell}}$) which is the root-mean-square distance of all the pixels in the mask with respect to the centroid. The average values of PI and the deviation as standard error in the mean were used to represent global changes in RNA localisation.

To calculate the overlapping fraction of foci from dually labelled smFISH experiments (Q670 and AF488 probes), images were first processed using the FISH-quant LoG filter followed by background subtraction. The overlapping fraction was then calculated using the ImageJ plugin, Just Another Colocalisation Plugin (JACoP)[58].

**Reporting summary.** Further information on research design is available in the Nature Research Reporting Summary linked to this article.

## Data availability
The data that support the findings of this study are available from the corresponding author upon reasonable request.

## Code availability
The custom Matlab scripts used are available from the corresponding author upon request.

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

## Acknowledgements

The authors would like to thank Dirk Dormann and Chad Whilding from the MRC-London Institute of Medical Sciences microscopy facility, Prof. Jorge Ferrer and Dr. Anthony Beucher for providing the pLenti plasmid and members of the Unrau lab for synthesising TO1-Biotin. The Single Molecule Imaging Group is funded by a core grant of the MRC-London Institute of Medical Sciences (UKRI MC-A658-5TY10), a Wellcome Trust Collaborative Grant (206292/Z/17/Z) and a Leverhulme Trust Grant (RPG-2016-214). PJU acknowledges NSERC Canada Discovery grant.

## Author contributions

A.D.C., P.J.U. and D.S.R. designed the studies. A.D.C. performed the experiments and analysed the data. A.D.C. and D.S.R. wrote the manuscript with input from P.J.U.

## Competing interests

The authors declare no competing interests.
