## [Peer Review File · Nature Communications]

REVIEWERS' COMMENTS:

Reviewer #1 (Remarks to the Author):

The authors implemented most of the comments which I and the other reviewers provided in the previous cycle of peer-review to Nat Methods.

I am now satisfied with the manuscript and think it should be accepted for publication. The authors made a convincing case that they achieved single-molecule labeling, which allows fixed & live cell imaging of single mRNA molecules.

This is the first paper that convinced me that RNA aptamers can be used for single-molecule imaging of mRNA and, frankly, I am eager to try it for myself.

One minor comment was not addressed (minor comment 8):

The authors should write in the methods - and not acknowledgments - section from where they obtained TO1-biotin (and write at least once the full name of the molecule). It will be helpful for future users if the authors can point out commercially available TO1-biotin (if available), since I think this will be the bottleneck of using this system.

Gal Haimovich

Reviewer #2 (Remarks to the Author):

The author's have addressed my major concerns. In particular, I appreciate the effort the author's have taken to improve the newer movies and images, which are more convincing. Below are my specific responses to each point brought up in my earlier review:

1. Co-movement now looks better and more believable in Fig. 3a and sup videos 3-4. I appreciate the effort to make better movies.
2. Looks much better.
3. OK
4. OK
- 5A. New movies look much better
- 5B. OK
6. Not focusing on transcription spot... ok
7. OK, processing fine.
8. OK, nucleolar signal low Fig. 2b and S2 now changed
9. Fig 2 subpanels still would be nice to show gray scale and then merge (perhaps with specific spots highlighted). Grayscale has the most contrast for the eye and is fairest.
10. OK, nice to mention that background may vary from one cell type to another.
11. I like the separate distributions for background. OK
12. OK
13. OK

Reviewer #3 (Remarks to the Author):

This study describes the development of a stable Mango II tag comprising three stem-modified Mango II aptamers. An M2x24 tag (containing 24 copies of the Mango II aptamer) was characterized in single molecule imaging, cellular localization, and super-resolution imaging in comparison to the MS2x24 tag that uses MCP-EGFP. A stand-out feature of the work is the variety of imaging experiments that address specific applications (single molecule imaging, cellular localization, super-resolution imaging) and known issues (cellular localization and translation). Several methodological advancements were required to use the M2x24 tag, such as low TO1-biotin concentration and aqueous imaging buffer. With the revisions to address prior reviewer critiques the results presented are of good quality and

will be of high interest to the imaging field as well as to the fields of RNA biology and synthetic biology / chemical biology.

Response to referees

Reviewer #1 (Remarks to the Author):

The authors implemented most of the comments which I and the other reviewers provided in the previous cycle of peer-review to Nat Methods. I am now satisfied with the manuscript and think it should be accepted for publication. The authors made a convincing case that they achieved single-molecule labeling, which allows fixed & live cell imaging of single mRNA molecules. This is the first paper that convinced me that RNA aptamers can be used for single-molecule imaging of mRNA and, frankly, I am eager to try it for myself.

One minor comment was not addressed (minor comment 8):

The authors should write in the methods - and not acknowledgments - section from where they obtained TO1-biotin (and write at least once the full name of the molecule). It will be helpful for future users if the authors can point out commercially available TO1-biotin (if available), since i think this will be the bottleneck of using this system.

Gal Haimovich

We are glad reviewer #1 is satisfied with our revisions. As previously suggested, we did include the source and full name of TO1-Biotin as it appears from Applied Biological Molecules inc. see quote from methods section of the revised manuscript. "The modified Thiazole Orange dye (TO1-3PEG-Biotin) was synthesized by Applied Biological Materials Inc." – This can be found in fluorescence measurements of mango arrays section. We apologize that this may not have been easy to locate previously.

Reviewer #2 (Remarks to the Author):

The author's have addressed my major concerns. In particular, I appreciate the effort the author's have taken to improve the newer movies and images, which are more convincing. Below are my specific responses to each point brought up in my earlier review:

1. Co-movement now looks better and more believable in Fig. 3a and sup videos 3-4. I appreciate the effort to make better movies.
2. Looks much better.
3. OK
4. OK
- 5A. New movies look much better
- 5B. OK
6. Not focusing on transcription spot... ok
7. OK, processing fine.
8. OK, nucleolar signal low Fig. 2b and S2 now changed
9. Fig 2 subpanels still would be nice to show gray scale and then merge (perhaps with specific spots highlighted). Grayscale has the most contrast for the eye and is fairest.
10. OK, nice to mention that background may vary from one cell type to another.

11. I like the separate distributions for background. OK
12. OK
13. OK

We thank the review for their comments and are glad the new data satisfies their concerns. Regarding point 9, we feel that adding grey scale images, although clearer to print, may detract from the continuity of the colour scheme used and would similarly add extra panels in an already busy figure. But we thank the reviewer for their comment.

Reviewer #3 (Remarks to the Author):

This study describes the development of a stable Mango II tag comprising three stem-modified Mango II aptamers. An M2x24 tag (containing 24 copies of the Mango II aptamer) was characterized in single molecule imaging, cellular localization, and super-resolution imaging in comparison to the MS2x24 tag that uses MCP-EGFP. A stand-out feature of the work is the variety of imaging experiments that address specific applications (single molecule imaging, cellular localization, super-resolution imaging) and known issues (cellular localization and translation). Several methodological advancements were required to use the M2x24 tag, such as low T01-biotin concentration and aqueous imaging buffer. With the revisions to address prior reviewer critiques the results presented are of good quality and will be of high interest to the imaging field as well as to the fields of RNA biology and synthetic biology / chemical biology.

We thank the reviewer for their comments and review.